# An Occurrence Study of Mycotoxins in Plant-Based Beverages Using Liquid Chromatography–Mass Spectrometry

**DOI:** 10.3390/toxins16010053

**Published:** 2024-01-17

**Authors:** Romans Pavlenko, Zane Berzina, Ingars Reinholds, Elena Bartkiene, Vadims Bartkevics

**Affiliations:** 1Institute of Food Safety, Animal Health and Environment “BIOR”, Lejupes Iela 3, LV-1076 Riga, Latvia; romans.pavlenko@bior.lv (R.P.); zane.berzina@bior.lv (Z.B.); vadims.bartkevics@bior.lv (V.B.); 2Department of Food Safety and Quality, Lithuanian University of Health Sciences, Tilzes Street 18, LT-47181 Kaunas, Lithuania; elena.bartkiene@lsmu.lt

**Keywords:** mycotoxins, plant beverages, liquid chromatography, mass spectrometry, thermal treatment, exposure assessment

## Abstract

Mycotoxins are toxic mold metabolites that can adversely affect human and animal health. More than 400 mycotoxins have been identified so far. Cereals and nuts are the predominant mycotoxin-contaminated foodstuffs. Plant-based drinks produced from cereals, nuts, and legumes have grown in popularity. The mycotoxins accumulated in these crops may transfer to these beverages. A liquid chromatography–tandem mass spectrometry method was developed and optimized for the assessment of 22 mycotoxins in commercially available plant-based drinks in Latvia and Lithuania. A total of 64% of the seventy-two analyzed beverages were positive for one to sixteen mycotoxins, with deoxynivalenol, beauvericin, and enniatins A, B, B1, T-2, and HT-2 toxins detected most frequently. The European Commission has not yet set guidelines for the maximum mycotoxin concentrations in plant-based beverages, nor has the European Food Safety Authority conducted a risk assessment. Therefore, acute exposure studies were provided for the Latvian population based on the assumed replacement of dairy milk with plant-based beverages to ascertain the safety of plant-based milk substitutes. Based on the observed levels of mycotoxin prevalence and contamination levels and assumed exposure, it can be concluded that tested plant-based beverages may be relatively safe. However, exposure to emerging mycotoxins should be considered.

## 1. Introduction

The global consumption of plant-based beverages has expanded over recent years [1]. Consumers choose these beverages as alternatives to dairy milk due to health, environmental, and lifestyle preference reasons. Plant-based beverages that are also known as non-dairy milk substitutes are popular for their taste, nutrient value, including vitamins, low-fat content, and ethical advantage, as well as related dietary and environmental aspects [2,3]. Investments are currently being made toward the further development and production of such products in order to refine the technology and improve the nutritional value of plant-based drinks, as well as improve their taste and extend their shelf-life [1]. At the same time, it is important to ensure the safety of these dairy milk alternatives. Plant-based drinks are considered environmentally friendly because of reduced carbon footprint and greenhouse gas emissions, and they also use less land use than the production of conventional dairy milk [4,5,6]. Plant-based food production emits four times less carbon dioxide (CO_2_-eq. per kg of product) on average than the production of dairy milk [6].

Mycotoxins are secondary metabolites of fungi that can be dangerous to human and animal health. The most common mycotoxins are aflatoxins (AFs), trichothecenes, fumonisins (FB1, FB2), zearalenone (ZEA), and ochratoxin A (OTA) [7,8]. Raw agricultural products, including cereals, legumes, nuts, fruits, herbs, and other crops used in herbal beverages, can be contaminated with fungi, leading to the occurrence of mycotoxins in processed foods, such as plant-based beverages [9,10,11]. Mycotoxins exhibit acute and chronic toxicity, including genotoxicity, carcinogenicity, immunotoxicity, mutagenicity, nephrotoxicity, and teratogenicity [7,8,12]. Mycotoxins are, therefore, harmful to health when consumed in significant quantities or continuously ingested [13].

*Fusarium* is a widespread pathogen affecting cereals, animal feed, and food products worldwide [14]. Under favorable conditions, the metabolism of *Fusarium* species can produce mycotoxins of the hexadepsipeptide type, such as beauvericin (BEA) and enniatins (ENNs), such as ENNA, ENNA1, ENNB, and ENNB1, which are the most common grain contaminants in Europe [15,16].

Limited information is available in the scientific literature on the mycotoxin content of plant-based beverages [9,10,11].

The highest concentrations of mycotoxins were found in nut and oat beverages, particularly almond-derived beverages. The concentration of tentoxin ranged from 15 to 98 µg L^−1^, the concentration of ENNB ranged from 10 to 109 µg L^−1^, and the concentration of ENNB1 ranged from 6 to 60 µg L^−1^ in almond-based beverages. Conversely, in oat beverages, the highest concentrations were observed for tentoxin, ENNs, ZEA, and HT-2 toxin [9,10,11].

The authors of the aforementioned publications mostly reported low mycotoxin concentrations, but in a few cases, mycotoxin content reached 109 µg L^−1^. For ENN B, alternaria monomethyl ether (AME) and fumonisins were not found in those studies. A study by Hamed et al. on aflatoxins in plant-based beverages could not detect any aflatoxins above quantification levels [17].

The European Commission still does not have a clear view of the maximum permissible levels of mycotoxins in plant-based beverages, and no risk assessment has been carried out by the European Food Safety Authority [18].

The European Commission Regulation No. 2023/915 sets maximum levels for the main mycotoxins in similar food commodities, and the Commission Recommendation No. 2013/165 sets the indicative levels for T-2 and HT-2 toxin in cereals and cereal products [18,19]. Indicative levels of 200 and 50 μg kg^−1^ are recommended for the sum of T-2 and HT-2 toxins in oat bran and flakes and in other cereal milling products. For OTA, DON, ZEA, and AFB1, the maximum levels of 3.0 μg kg^−1^, 200 μg kg^−1^, 20 μg kg^−1^, and 2.0 μg kg^−1^ are indicated in the products derived from unprocessed cereal grains and processed cereal foods. The maximum levels for AFB1 and the sum of aflatoxins in oil seeds (such as soybeans, palm nuts, and hemp seeds) and tree nuts (including cashew nuts and cocoa beans) are 2.0 μg kg^−1^ and 4.0 μg kg^−1^, respectively. For AFB1 in almonds, hazelnuts, and Brazil nuts intended for use as ingredients in food, the maximum levels are between 8 μg kg^−1^ and 12 μg kg^−1^ [18,19].

The aim of the present study was to investigate the prevalence of mycotoxins in plant-based beverages, develop and validate a method for mycotoxin detection, and perform a consumer exposure assessment.

In this study, an analytical method using QuEChERS extraction and ultra-high-performance liquid chromatography with tandem mass spectrometry detection (UHPLC-MS/MS) was used for the analysis of 22 mycotoxins in 72 plant-based milk substitutes.

An exposure assessment study was provided based on the assumption of plant-based beverages as a replacement for dairy milk in the daily diet.

In this study, low quantification levels were achieved for this type of matrix, and new compounds (NIV, TEN, 3-ADON, and 15-ADON) not previously studied in plant-based beverages were investigated and new matrices, such as hemp, millet, buckwheat, and pea beverages, have been included in this study.

## 2. Results and Discussion

### 2.1. Optimization of the Sample Preparation

#### 2.1.1. Comparison of the SPE and QuEChERS Methodologies

Two sample preparation methods were compared, namely, the “quick, easy, cheap, effective, rugged, and safe” (QuEChERS) method and C18 column-based solid phase extraction (SPE) for the evaluation of the extraction efficiency (EE). The EE was calculated as described in Section 4.2.5.

The spiking levels used during the validation studies were 0.031 μg kg^−1^ for aflatoxins AFB2 and AFBG2; 0.125 μg kg^−1^ for aflatoxins AFB1 and AFG1; 0.5 μg kg^−1^ for OTA; 0.7 μg kg^−1^ for ZEA; 1 μg kg^−1^ for ENNA, ENNA1, ENNB, ENNB1, and BEA; 1.25 μg kg^−1^ for tentoxin (TEN), T-2, and HT-2 toxins; 2 μg kg^−1^ for alternariol (AOH) and AME; 2.5 μg kg^−1^ for DON; 4 μg kg^−1^ for FB2; 15 μg kg^−1^ for nivalenol (NIV); 20 μg kg^−1^ for 3-acetyldeoxynivalenol (3-ADON) and 15-acetyldeoxynivalenol (15-ADON); and 1000 μg kg^−1^ for FB1.

The extraction efficiencies of both the SPE and C-18 QuEChERS methods were compared based on the EE assessment obtained for eighteen major mycotoxins assessed in the soy matrix in three replicates (see Figure 1). 

For most of the mycotoxins, the EE was higher using the QuEChERS methodology, with the exception of HT-2 toxin and fumonisin B1, for which efficiency was higher in the case of the SPE procedure. A study by Berzina et al. reported that when using SPE C18-E columns, deoxynivalenol (DON) had poor absolute recoveries due to the weak bonding to the C18 sorbent [20]. Nivalenol (NIV) is a cooccurring *Fusarium* metabolite similar in structure and properties to DON, which also presented poor retention on C18 sorbent. The same study described a difficulty in eluting BEA from the SPE C18-E column due to the non-polar nature of this analyte, resulting in strong binding to the sorbent, thus requiring an increased eluent volume [20]. The QuEChERS method resulted in low analyte signals for the BEA and ENN groups because of inefficient extraction to the organic layer [21].

Fumonisins and OTA were irreversibly adsorbed using PSA, as described in other publications [21], because of the strong ionic affinity between the primary and/or secondary amines in the PSA carrier and the carboxyl groups present in the mycotoxin [22]. For the other mycotoxins, extraction efficiency did not change significantly. It was decided not to use PSA salts in order to allow for effective analysis of acidic mycotoxins.

Based on the conducted experiments, it was decided to perform the analysis of real samples using a commercially available QuEChERS salt mixture due to its simplicity and comparably high extraction efficiency.

#### 2.1.2. Selection of a Reconstitution Solution and Sample Filtration

In order to reduce the matrix effects and save time, the possibility of concentrating 1 mL of extract was tested instead of 4 mL extract volume as proposed in a former study by Reinholds et al. [23]. The application of 1 mL evaporated extract caused a significant reduction in the sensitivity and hindered the detection of low mycotoxin concentrations around the level of quantification (LOQ).

The reconstitution solution with high water content is more suitable for the detection of polar compounds, such as DON and NIV. On the other hand, for non-polar compounds, such as BEA and the ENN group, a reconstitution solution with an increased organic solvent content is preferable. Using a solution with high water content reduced the recovery of non-polar compounds for chromatographic analysis. It was, therefore, necessary to find a reconstitution solution with the optimal solubility of both polar and non-polar mycotoxins that would also be acceptable for chromatographic separation. It was also necessary to adjust the injection volume to obtain a satisfactory signal shape.

To find the most effective reconstitution solution, aqueous 15%, 25%, and 50% methanol solutions with 0.5 mM ammonium acetate and 0.1% formic acid, an aqueous 50% methanol solution with 0.5 mM ammonium acetate and 1% formic acid, and aqueous 15% and 25% acetonitrile solutions with 0.5 mM ammonium acetate and 0.1% formic acid were compared. The experiment was prepared in three replicates.

When the organic solvent content exceeded 25% methanol or 15% acetonitrile, asymmetric signal shapes and double signal formations were observed for polar compounds, while enhanced signal intensity was observed for non-polar compounds, like BEA and ENNs. The optimal signal shapes and intensities were obtained by reconstructing the samples in an aqueous 50% methanol solution supplemented with 0.5 mM ammonium acetate and 1% formic acid. When using this reconstitution solution, DON and NIV signal shapes were not optimal, but this fact did not affect the quantification of those signals.

The selected reconstitution solution did not match the initial mobile phase composition; therefore, the injection volumes were tested, and 5 µL was chosen to achieve the optimal signal shape for DON and NIV.

Some mycotoxins were absorbed on polyvinylidene difluoride (PVDF) centrifuge filters, and thus freezing and centrifugation were applied for the final purification of the extract.

### 2.2. Validation of the Method

The proposed method for mycotoxin analysis based on UHPLC-MS/MS is described in Section 4.2. and has been used for the assessment of 22 selected mycotoxins. The method was validated using four different types of plant-based (e.g., oat, rice, soya, and almond) beverages according to the current recommendations based on the European Commission Regulation (EC) No. 401/2006 and SANTE 12089/2016 “Guidance on the identification of mycotoxins in food and feed” [24]. The method characteristics, including recovery, repeatability, selectivity, linearity, method sensitivity, and matrix effects, were evaluated (see Appendix A). The calculations for the determination of matrix effect (ME) and the method uncertainty are summarized in Section 4.2.5.

#### 2.2.1. Levels of Quantification, Method Selectivity, and Linearity

The selectivity of the method was tested by analyzing blanks and samples spiked with a standard at the LOQ (see Table 1).

The retention time difference between the signals in the spiked sample and the signals in the standard solution was less than 1%. No interfering signals were observed in the chromatograms. The method showed good selectivity.

The linearity and correlation coefficients (R^2^ > 0.98) of the method were tested by a five-point procedural calibration ranging from the LOQ level to 40× LOQ level. Good linearity was maintained over the selected concentration range, with the obtained correlation coefficients being at least ≥0.98.

The level of quantification (LOQ) values (Table 1) were expressed as the concentrations of mycotoxins corresponding to a signal-to-noise value (S/N) ≥ 10 [25]. One of the lowest LOQ values compared to other publications was obtained in the present study.

In a paper by Hamed et al., Afs LOQ values were estimated to be 0.5 µg L^−1^ [17], whereas in the present study, the LOQ values for the Afs were 0.0078–0.063 µg kg^−1^. The LOQ values for the ENNs and BEA obtained by Arroyo-Manzanares et al. were 0.3–0.8 µg L^−1^, depending on the type of the plant-based beverage [11]. In the present study, the LOQ values for BEA and the ENNS were three to eight times lower (0.10 µg kg^−1^) for all of these mycotoxins in all types of tested beverages. The LOQ values for all mycotoxins described by Miro-Abella et al. (2017) ranged from 0.05 to 15 µg L^−1^ [9], while the LOQ values in the study by Juan et al. ranged from 0.3 to 18 µg L^−1^ [10].

#### 2.2.2. Repeatability and Recovery

The method repeatability and recovery were tested for each matrix sample spiked with a mycotoxin standard at the LOQ level and at 2–10 times the LOQ level.

In addition, for all beverages, the two main concentration levels tested were the LOQ and 5× LOQ levels in five replicates for each level (the number of replicates was different for the aflatoxnis because four aflatoxins were in the mixture). Additionally, data from quality control experiments (spiked samples) for the calculation of method performance characteristics (for example, uncertainty) were used.

Each sample series included at least one quality control sample for each matrix. Detailed recovery values for mycotoxins are given in the Appendix A. The average mycotoxin recoveries for all matrices are given in Table 2.

#### 2.2.3. Matrix Effect and Extraction Efficiency

The matrix effects (MEs, %) were tested by calibrating the standard solution and calibrating the matrix by adding mycotoxin standards to the extracts during the evaporation step. Calibration with standard solutions cannot be applied to the quantification of mycotoxins in real samples due to matrix effects. In order to compensate for matrix effects, it is necessary to use an internal standard or matrix-matched calibration (Figure 2).

The matrix effects were the highest for soy drinks, except for the enniatin group, BEA, OTA, and fumonisins. In all matrices, the matrix effects for fumonisins were positive, suggesting ion enhancement.

The EE was tested for all types of plant-based beverages in three replicates. For most mycotoxins, the EE was greater than 50%. In the case of oat and rice beverages, reduced extraction efficiencies were obtained for BEA and ENNA1, probably due to the applied reconstitution solution. However, for DON and NIV, a reconstitution solution containing more than 50% organic solvent was not suitable, and SPE columns, such as C18-E, were unable to retain DON and NIV due to the polarity of these analytes.

### 2.3. Method Application

The developed methodology was applied to the analysis of plant-based beverages obtained from the retail market. The results are shown in Appendix A.

Aflatoxins (except for AFB2), fumonisins, NIV, 3-ADON, and 15-ADON were not observed in oat beverages. The most frequently detected mycotoxins were ENNB, ENNB1, DON, BEA, ENNA, HT-2, and T-2-toxin. ENNB and ENNB1 were detected in 20 samples. The mycotoxin burden in oat drinks was dominated by a high percentage of ENN B and ENN B1, which was in agreement with the publication by Juan et al. (2022) [10].

The highest concentrations were found for DON, ENNB1, AME, and TEN (9 ± 2, 3.2 ± 0.8, 3.0 ± 0.4, and 2.7 ± 0.5 µg kg^−1^, respectively).

Only two mycotoxins were detected in nut beverages: AFB1 and AFB2. AFB1 was detected at the highest concentration of 0.07 ± 0.02 μg kg^−1^.

The paper by Juan et al. mentioned other mycotoxins in almond drinks, such as ENNs, whereas the occurrence of AFs in this type of drink was only determined in 5% of tested samples [10]. In this study, no other mycotoxins were detected above the LOQ value in nut drinks. However, traces of AFB1 and AFB2 were detected at very low levels. Other authors have also reported low levels of AFs in nut-based beverages [17].

Eleven mycotoxins (DON, AOH, BEA, ENN A, ENB, ENNB1, ZEA, T-2, TEN, AFB1, and AFB2) were detected in rice beverages. In general, rice beverages have relatively low mycotoxin content, according to other publications [9,10,11,17].

In this study, ENNs, DON, AFB1, and AFB2 are frequently found in rice beverages. Juan et al. also mentioned that the enniatin group is predominant in rice beverages.

In the present study, DON, ZEA, ENNB, and ENNB1 were detected in soy beverages at low concentrations. The highest concentration was revealed for DON at 2.0 ± 0.3 μg kg^−1^.

A total of ten mycotoxins were detected above the LOQ value in other tested plant- beverages based on hemp, millet, buckwheat, and peas. The most frequently detected among these mycotoxins were ENN B1, OTA, and TEN. The highest concentrations were quantified for NIV in the buckwheat drink, for AOH in the pea drink, and for TEN in the hemp drink. Prior to our study, there was no data in the literature on the occurrence of mycotoxins in hemp, millet, buckwheat, and pea-based beverages.

### 2.4. Results of Thermal Stability Testing

The results of thermal processing tests for the plant beverages are presented in Figure 3 and Figure 4.

Mycotoxin concentrations decreased with heat treatment, except for AFB1. In the almond drink, AFB1 concentrations slightly increased during heating.

The heating of plant-based beverages in a coffee machine reduced mycotoxin concentrations more than boiling, despite the fact that the treatment temperature in the coffee machine simulation was lower compared to the boiling simulation. This was probably caused by the fact that the beverage heated up much faster in the coffee machine than during boiling. The steam temperature in the coffee machine can reach 130 °C, according to the user manual. This could have a major effect on the reduction in mycotoxin content.

In order to correctly assess the thermal treatment effects on mycotoxin concentrations, the resulting mycotoxin concentrations must be outside the uncertainty limit of the compound analysis. Otherwise, the apparent changes in mycotoxin concentrations may be caused by random fluctuations. The change in mycotoxin concentration after thermal treatment exceeded the method uncertainty for the oat drink.

The boiling of the oat drink resulted in an 8–18% reduction in mycotoxin concentrations. In contrast, processing the same drink in a coffee machine resulted in a 49–100% reduction in mycotoxin concentrations.

### 2.5. Exposure Risk Studies

#### 2.5.1. Consumption of Milk in Latvia

Dietary exposure to mycotoxins was assessed based on the available data on mycotoxin contamination in beverages and the estimated acute intake data. First, the Comprehensive Food Consumption database of the European Food Safety Authority (EFSA) was checked to find some data regarding the consumption of non-dairy milk substitutes in Latvia and other Baltic states. However, there was no data available regarding the Latvian population. Thus, it was decided to use recent data from a national survey by Siksna et al. (2020) for salt and iodine consumption in the adult population in Latvia, which also presented the average consumption data over a three-day period for different milk products, including pasteurized dairy milk, among adults in Latvia [26].

These data are summarized in Table 3. Due to the absence of direct data on plant-based milk substitute consumption, we elected to use data on dairy milk intake due to the assumption that certain populations may fully replace their dairy milk intake with plant-based non-dairy alternatives. The provided prediction of dairy milk replacement allows us to characterize at least the potential worst-case exposure to plant-based milk substitute beverages, as well as anticipate the likely future trends in this direction.

It must be emphasized that there was a 1.7 to 1.5 times lower milk consumption rate among females compared to males. Furthermore, the milk consumption level among males decreased by almost two times with an increase in age, whereas for females, there was little difference in age, as the consumption rate was almost the same. This may be attributed to the cultural habits of older people, as plant-based beverages are still not very popular among elderly people.

It was indicated from the data that male toddlers and younger males between 19 and 34 years were the major groups that could be exposed to mycotoxin contamination by replacing milk with plant-based milk alternatives.

#### 2.5.2. Exposure Assessment

Section 4.4 describes the methodology used for the exposure estimation based on the assumption of conventional milk replacement with plant beverages.

Many studies have been dedicated to mycotoxin exposure assessment via the consumption of teas, as well as fruit and vegetable juices, but there is still a lack of data about human exposure from plant-based milk substitute beverages. Considering that plant-based milk substitutes are mostly used in addition to tea and coffee, the PDI values were calculated based on this assumption. The calculated PDI values for males and females based on the upper bound (UB) average values of contamination and the highest concentration levels determined in the studied plant-based beverages are summarized in Table 4.

The calculated PDI values for the average UB mycotoxin contamination levels ranged between 0.01 and 7.62 ng kg^−1^ body weight per day for all age groups of males, whereas the highest levels were determined for the exposure of 19–39 years old males to NIV, and the lowest value was determined for AFB1 based on the average contamination levels in nut-based beverages. For females, the PDI for the average UB levels ranged between 0.01 and 4.29 ng kg^−1^ body weight per day, and the highest levels were found in 19- to 34-year-old females in the case of exposure to NIV. It must be emphasized that compared to the studies by other authors on the overall mycotoxin exposure rates in the European population, our study revealed PDI values that were quite low for most of the EU-regulated mycotoxins. A recent study of PDI outputs simulated for adults in Northern Europe indicated the average UB levels of the PDI for OTA, DON, and ZEN to be 0.6764, 0.48282, and 0.00149 µg kg^−1^ body weight per day, respectively [27]. It can be expected that the determined levels of mycotoxin contamination in plant-based milk substitute beverages are very low compared to these simulated values.

It can be pointed out that compared to males, the PDI data for all age groups of females did not exhibit major differences between the groups.

The PDI values of the UB contamination levels were used for the calculation of the potential exposure (%TDI values) that are shown in Table 5.

For the commonly found Fusarium mycotoxins, such as DON and ZEA, as well as the emerging mycotoxins (a sum of ENNs, BEA, and TEN), the values of determined PDI were shown to be below 1% of the determined TDI or threshold of the concern, considering the determined average UB levels. For NIV and AFB2, the levels were slightly above 1% of the recommended TDI by the EFSA.

The %TDI values for the sum of T-2 and HT-2 toxins were almost two times higher for males in the age group of 19–34 years compared to females of the same age, whereas the average values for all age groups ranged between 4.66 and 6.84%. It was determined that for the carcinogenic mycotoxins, such as AFB1, the sum of determined AFs, OTA, the %TDI values were also below 10%, indicating quite low risks. While the %TDI values for AOH and AME were quite high and reached 35.1 to 43.5% of the actual exposure levels for males and 1.5 times lower values (29.7 and 23.9%) for females, it must be emphasized that these mycotoxins are considered to have low acute toxicity, whereas their chronic effects can be quite severe. as these mycotoxins are reported to cause esophageal cancer [28].

Overall, the UB level studies indicated that all the studied beverages are safe for human consumption. However, it was also evaluated whether the highest concentration levels determined in the tested beverages may have some impact on the %TDI. Similar to the UB levels, the PDI values were also calculated for the maximum concentration levels (data summarized in Table 6).

It can be seen that DON and its derivative NIV had the highest PDI levels compared to other mycotoxins due to their higher concentrations. For example, the PDIO ranged between 10.5 and 7.1 ng kg^−1^ body weight per day for DON when comparing the data for all male and female age groups. It must be emphasized that if we divide the PDI values at the maximum contamination levels and the UB average data assessment for the younger age group or even for all age groups of males, then the ratio of PDI at the maximum mycotoxin concentration is 5 to 7 times higher compared to the average contamination levels considering the UB approach. For example, for DON, TEN, the sum of ENNs, OTA, and AME, the PDI (max) values were 6.9, 6.6, 5.0, 4.8, and 4.2 times higher compared to the PDI values for the average contamination levels for all male age groups. The same tendency can be seen in the case of the females. Obviously, this indicates the notable impact of contamination levels on the probable impact of mycotoxins during daily consumption. Even for the other tested mycotoxins, the PDI values considering the highest mycotoxin concentrations were about 1.5 to 2 times higher compared to the average mycotoxin concentration, indicating that the frequency of mycotoxin contamination and its prevalence was very different in the tested beverages.

This can be illustrated by comparing the %TDI values and taking into consideration the highest mycotoxin contamination levels determined in the tested plant beverages (Table 7).

Despite the relatively lower %TDI for females, we can see a rather important increase for the male subgroup at the age of 19–34. The %TDI for AFB1 and the sum of the two determined AFs were 10.5% and 13.5%, respectively. It can be seen that in the case of OTA, this ratio is even five times higher than %TDI for the OTA average levels and is also very high for the sum of T-2 and HT-2 toxins (24%) in this subgroup of younger males. It must be emphasized that this group may have an even higher risk in the case of an additional use of plant beverages as additives to cereals or other food products, not only by adding them to hot drinks. For the female groups, these values are up to two times lower but still present some concern about the impact of mycotoxins.

As noted previously, AOH and AME may be more relevant to the long-term exposure risks, whereas the %TDI exceeded the threshold of toxicological concern (TTC), indicating that these mycotoxins may present actual risks. Obviously, more studies about their presence are needed.

## 3. Conclusions

An analytical methodology based on QuEChERS extraction and HPLC-MS/MS was developed for the analysis of 22 mycotoxins in plant-based milk substitute beverages. The method parameters (correlation coefficient ≥ 0.98, LOQ 0.0078–5.0 μg kg^−1^, method recovery 79–119%) complied with the European Union Commission Regulation No. 401/2006 [24]. A total of sixteen mycotoxins were detected in the tested beverages, and 61% of all analyzed beverages contained at least one mycotoxin. The most frequently detected mycotoxins were enniatin B1 in soy drinks, enniatins B and B1, deoxynivalenol, beauvericin, enniatin A, HT-2 and T-2 toxins in oat drinks, aflatoxin B2 in nut drinks, deoxynivalenol, enniatins B and B1, aflatoxins B1 and B2 in rice drinks, and enniatin B1 in hemp, millet, buckwheat, and pea beverages.

The oat drink was one of the most contaminated plant-based drinks, with the highest detected deoxynivalenol concentration at 9 ± 2 μg kg^−1^. Thermal treatment of plant beverages did not consistently reduce mycotoxin concentrations, with several mycotoxins remaining unchanged. The highest reduction of 49–100% was observed when the oat drink was processed in a coffee machine. This study found that mycotoxin contamination in plant-based beverages generally poses a low acute health risk for the Latvian population. Most mycotoxin exposure levels were well below the recommended thresholds. However, exposure to AOH and AME may require further investigation due to potential chronic health concerns. Additionally, higher mycotoxin exposure in specific population groups, especially young males, could lead to elevated risks. Overall, plant-based milk substitute beverages can be considered safe, but further research is needed to fully understand the long-term effects and potential risks associated with mycotoxin exposure.

## 4. Materials and Methods

### 4.1. Samples of Plant-Based Beverages

Seventy-two different plant-based beverages were purchased from retail stores in Riga (Latvia) and Kaunas (Lithuania), including oat, soy, nuts, rice, hemp, millet, buckwheat, and pea beverages. Nut beverages were made from almonds, coconuts, Brazil nuts, and cashews. The selected samples were as diverse as possible in order to characterize the differences in mycotoxin content. Some herbal drinks contained added flavors such as chocolate and vanilla, supplemental vitamins, and extra sugar. Plant-based beverages intended for coffee brewing were also sampled. The samples were stored at room temperature (+20 °C) before opening the packages, followed by placing the samples in a freezer at −20 °C until the analysis.

### 4.2. Mycotoxin Analysis

#### 4.2.1. Standards and Reagents

Mycotoxin standards were of at least 95% purity. Standards of AME, BEA, NIV, and four ENNs, A, A1, B, and B1, were purchased from Cayman Chemical Company (Ann Arbor, MI, USA). Standards of DON, FB1 FB2, HT-2 and T-2 toxins, OTA, ZEA, 3-ADON and 15-ADON, TEN, and a mycotoxin mix containing four aflatoxins (AFB1, AFB2, AFG1, and AFG2) were purchased from Biopure (Guntramsdorf, Austria).

OTA standard was shipped in acetonitrile at 10 µg mL^−1^. A mycotoxin mix containing four aflatoxins was shipped in acetonitrile at 2.0 µg mL^−1^ for AFB1 and AFG1 and 0.50 µg mL^−1^ for AFB2 and AFG2.

Formic acid (99%) was obtained from VWR Chemicals (Radnor, PA, USA). Methanol (LC grade), N,N-dimethylformamide (DMF), and acetonitrile were purchased from Merck (Darmstadt, Germany). Ammonium acetate (≥99%) was supplied by Ing. Petr Švec—PENTA (Prague, Czech Republic). Deionized water (18.2 MΩ × cm) was further purified with a Mili-Q-gradient academic A10 water purification system (Millipore, Darmstadt, Germany). QuEChERS buffer-salt extraction kits consisting of magnesium sulfate (4 g), sodium chloride (1 g), trisodium citrate dihydrate (1 g), and disodium hydrogen citrate sesquihydrate (0.5 g) per portion were purchased from Phenomenex (Torrance, CA, USA).

#### 4.2.2. Preparation of Standard Solutions

Stock standard solutions were prepared by dissolving solid standards in acetonitrile. Beauvericin solution was obtained by dissolving in DMF. The stock solutions had concentrations ranging from 300 to 2012 µg mL^−1^. Working standard solutions were prepared in acetonitrile.

#### 4.2.3. Sample Preparation

Samples of plant-based beverages (10.00 ± 0.01 g) were weighed into 50 mL polypropylene tubes (Sarstedt, Nümbrecht, Germany). The mycotoxin working solution was added to calibration samples, producing 5 calibration points ranging from the LOQ to 20 × LOQ and quality control samples as real samples with standard addition. The samples were extracted in 10 mL of acetonitrile containing 0.2% formic acid and shaken for 10 min using an overhead shaker (Biosan, Riga, Latvia). The QuEChERS salt mixture was added to each extract followed by shaking for 10 min using an overhead shaker. The tubes were centrifuged (Thermo Fisher Scientific, Waltham, MA, USA) at 4500 rpm for 10 min. The supernatant was placed in a freezer (Thermo Fisher Scientific, Waltham, MA, USA) at −80 °C for 30 min, and 4 mL of the supernatant after centrifugation for 15 min at 4500 rpm at 20 °C was evaporated at 50 °C and dissolved by vortexing (IKA, Staufen, Germany) in 300 µL of 50% aqueous methanol containing 1% formic acid and 0.5 mM ammonium acetate. The final extracts were transferred to Eppendorf tubes (Sarstedt, Nümbrecht, Germany) and frozen at −80 °C for 15 min. The supernatants after centrifuging at 12,000 rpm for 5 min at 20 °C were transferred into glass vials (Aijiren, Quzhou, China) for LC-MS/MS analysis.

#### 4.2.4. UHPLC–MS/MS Conditions

The analysis was performed on an UltiMate 3000 (Thermo Fisher Scientific, Waltham, MA, USA) HPLC system coupled with a TSQ Quantiva MS/MS detector (Thermo Fisher Scientific, Waltham, MA, USA). The mycotoxin separation was performed on a Kinetex C18 (Phenomenex, Torrance, CA, USA) reversed-phase analytical column (50 × 3.0 mm, 1.7 µm) at 40 °C using an injection volume of 5 µL. The mobile phase consisted of a 0.5 mM ammonium acetate solution containing 0.1% formic acid in Mili-Q water (eluent A) and a 0.5 mM ammonium acetate solution containing 0.1% formic acid in methanol (eluent B). A flow rate of 0.35 mL min^−1^ was used. The following gradient conditions were applied: 0.0 min, 15% B; 1.0 min, 15% B; 4.5 min, 98% B; 8.5 min, 98% B; 9.5 min, 15% B; and 15.0 min, 15% B.

Mycotoxin detection was conducted in positive and negative ESI modes. The MS parameters were as follows: needle voltage, +3500 V and −2500 V; sheath gas, 55 arbitrary units (Arbs); aux gas, 25 Arbs; sweep gas, 5 Arbs; ion transfer tube temperature, 300 °C; and vaporizer temperature, 350 °C. The main ion fragments were identified using SRM, and the scanning properties were as follows: cycle time, 0.3 s; Q1 resolution, 0.7 FWHM; Q3 resolution, 1.2 FWHM; CID gas, 1.5 mTorr; and chromatographic peak width, 6 s. The mycotoxin m/z transitions are summarized in Table 8.

#### 4.2.5. Extraction Efficiency, Matrix Effect, and Method Uncertainty Calculations

During the sample preparation optimization for QuEChERS extraction, extraction efficiencies were tested using salts of primary secondary amines (PSAs), a mixture of magnesium sulfate (4 g) and sodium chloride (1 g), and a mix of magnesium sulfate (4 g), sodium chloride (1 g), sodium citrate tribasic dihydrate (1 g), and sodium citrate dibasic sesquihydrate (0.5 g). Samples were prepared in three replicates.

The EE was tested as the ratio between the blank sample spiked with the standard after the procedure and the sample spiked with the standard at the beginning of the procedure for each compound (Equation (1)):(1)$$\%EE=\frac{A_{before}-A_{non\;spiked}}{A_{after}-A_{non\;spiked}}\times 100,$$
where *A_before_* is the area of the chromatographic peak for a spiked sample before the extraction procedure, *A_non spiked_* is the area of the chromatographic peak for a blank sample at the end of the sample preparation procedure, and *A_after_* is the area of the chromatographic peak for the blank sample spiked with the standard after sample preparation.

The matrix effect (*ME*, %) was estimated as the ratio of the slope coefficients of the calibration curves of matrix calibration and solution calibration (Equation (2)):(2)$$\%ME=\left(\frac{slope\;of\;procedural\;calibration\;curve\;on\;matrix}{slope\;of\;standard\;calibration\;curve}-1\right)\times 100,$$

If the matrix effect is between −20% and 20%, it is considered that this effect is negligible. A matrix effect that is lower than −20% corresponds with ion suppression, while a matrix effect that is higher than 20% corresponds with ion enhancement. The method uncertainty, *U_c_*, was calculated according to Equation (3):(3)$$U_{C}=2\cdot \sqrt{RSD^{2}+{\left(100\%-R\right)}^{2}}$$
where *R* is the recovery (%) and *RSD* is the relative standard deviation (%).

### 4.3. Thermal Stability Evaluation of Mycotoxins

The thermal stability of mycotoxins was tested in oat, rice, soy, and almond-based beverages. The thermal stability testing was carried out under two different conditions, simulating the boiling of plant-based beverages at 100 °C for 5 min and the use in a coffee machine (Jura, Niederbuchsiten, Switzerland) by heating at 60–70 °C for 10 s. All experiments were performed as 3 replicates for each matrix to assess the standard deviation of the results.

### 4.4. Exposure Study Methodology

Exposure assessment for the acute intake of mycotoxins was based on our previously reported approach applied to dairy products, albeit with some changes [23].

Two scenarios were used for assessing the impact of mycotoxin levels found in the beverages. The average upper bound (UB) contamination and the highest observed contamination level were both used for the calculation of probable daily intake values. Only the detected mycotoxins were selected for the assessment of exposure due to the daily consumption of milk, whereas for the calculation of the average UB levels, samples with levels below the LOQ were set to the numerical value of the LOQ, according to EFSA recommendations for the dietary exposure evaluation [29].

The consumption data were modeled based on the principle of dairy milk replacement with plant-based beverages, as there were no sufficient data regarding the actual intake of plant-based milk substitutes in Latvia and Lithuania where the samples were purchased.

The latest consumption data for dairy milk in Latvia were used for the calculation of the average acute daily intake, e.g., the probable daily intake (PDI) (µg kg^−1^ body weight per day), which was calculated according to the Equation (4):PDI = (C × M/1000)/W,(4)
where C is the determined maximum or the average upper bound level or the maximum mycotoxin concentration in (µg kg^−1^), M is the average acute consumption (g day^−1^), a unit of 1000 was used to convert g to kg, and W is the average body weight of 70 kg for adults and elderly according to the EFSA recommendations [30].

The probable acute exposure risk level, e.g., %TDI (µg kg^−1^ body weight per day), was calculated by dividing the PDI values by the tolerable daily intake (TDI) values, as shown by Equation (5):%TDI = (PDI/TDI) × 100,(5)
where %TDI is the estimated percentage of exposure, PDI is the calculated probable daily intake (µg kg^−1^ body weight per day), and TDI is the tolerable daily intake value (µg kg^−1^ body weight per day) determined by the authorities, such as the EFSA and the Joint FAO/WHO Expert Committee on Food Additives (JECFA).

In the case of mycotoxins lacking proper recommendations of the tolerable intake levels due to the uncertainty about the acute impact and predicted chronic toxicological effects, the other parameters, such as the threshold of toxicological concern (TTC) based on the EFSA recommendations, were used for calculating the exposure level [29,31].

The determined TDI, TCC, and CPF factors used for the calculation of the %TDI compared to the proposed PDI values based on dairy milk replacement with the tested plant-based beverages are summarized in Table 9 [32,33,34,35,36,37,38,39].

For aflatoxins as group I carcinogens, it is assumed that there is no threshold of exposure, and no TDI can be established. A recent study reported the cancer potency factor (CPF) proposed by the JECFA, which was set in the case of the dietary exposure level of AFB1 at 1 ng μg kg^−1^ bw per day. It was determined that the same factor can be attributed to the sum of AFs and was used during our study [37].

For OTA, The Heath Canada Organization suggested that 21 ng kg^−1^ bw per week (corresponding to TDI = 3 ng kg^−1^ bw per day) would be a more appropriate TWI value, and it is expected that this may be used as a replacement and was accepted as being appropriate instead of the 17 ng kg^−1^ bw per day (corresponding to 120 ng kg^−1^ bw per week), as previously established by the EFSA [39].

The determined TDI, TCC, and CPF factors used for the calculation of the %TDI compared to the proposed PDI values based on dairy milk replacement with the tested plant-based beverages are summarized in Table 9. The rather low levels of the parameters used for the calculations were expressed in ng kg^−1^ body weight per day (Table 9).

## Figures and Tables

**Figure 1 toxins-16-00053-f001:**
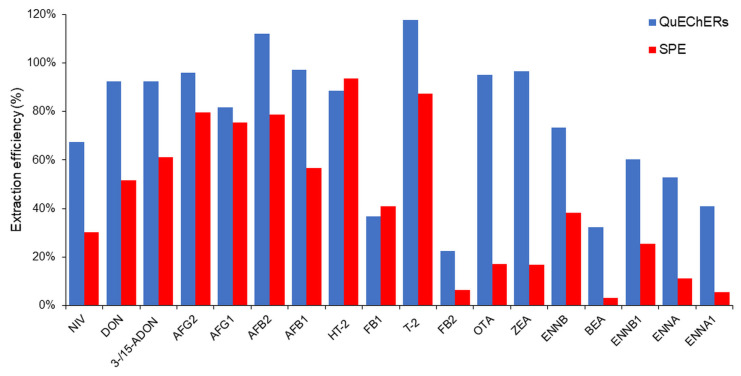
Comparison of QuEChERS and SPE C18 extraction efficiencies.

**Figure 2 toxins-16-00053-f002:**
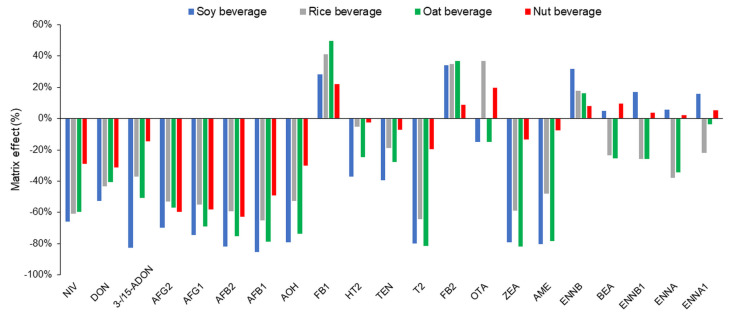
Matrix effects in the LC-MS-MS analysis of plant-based beverages.

**Figure 3 toxins-16-00053-f003:**
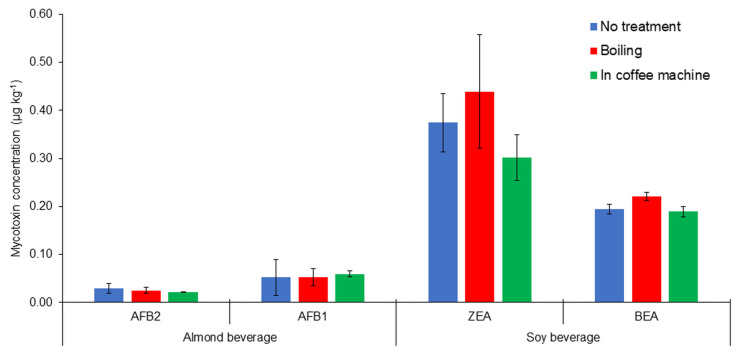
Thermal processing results for mycotoxins in almond and soy beverages.

**Figure 4 toxins-16-00053-f004:**
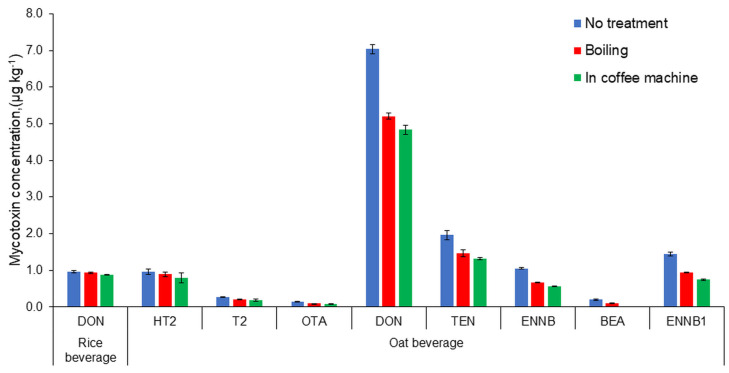
Thermal processing results for mycotoxins in rice and oat beverages.

**Table 1 toxins-16-00053-t001:** Levels of quantification (µg kg^−1^).

Compound	Matrix			
	Oat Beverage	Rice Beverage	Soya Beverage	Almond Beverage
ENNA	0.10	0.10	0.10	0.10
ENNA1	0.10	0.10	0.10	0.10
ENNB	0.10	0.10	0.10	0.10
ENNB1	0.10	0.10	0.10	0.10
BEA	0.10	0.10	0.10	0.10
DON	0.25	0.63	0.63	0.25
ZEA	0.18	0.18	0.18	0.18
FB1	1.0	1.0	1.0	1.0
FB2	0.25	0.25	0.25	0.25
NIV	5.0	5.0	5.0	5.0
OTA	0.050	0.050	0.050	0.050
SUM of 3ADON/15-ADON	2.0	5.0	5.0	2.0
T-2	0.13	0.31	0.31	0.13
HT-2	0.50	0.63	0.50	0.50
TEN	0.13	0.31	0.31	0.13
AFB1	0.031	0.031	0.063	0.031
AFG1	0.031	0.031	0.063	0.031
AFB2	0.0078	0.0078	0.016	0.0078
AFG2	0.031	0.031	0.031	0.0078
AOH	0.50	0.50	0.50	0.50
AME	0.50	0.50	0.50	0.50

**Table 2 toxins-16-00053-t002:** The average recovery values for mycotoxins.

Mycotoxins	Method Recovery (%), Average
DON	100
AFB1	101
AFB2	99
AFG1	100
AFG2	101
ZEA	100
HT-2	93
T-2	98
OTA	99
FB1	99
FB2	98
NIV	99
Sum of 3-ADON and 15-ADON	101
AOH	100
TEN	97
AME	101
ENNB	101
BEA	100
ENNB1	97
ENNA	98
ENNA1	97

**Table 3 toxins-16-00053-t003:** The average consumption data of dairy milk among the Latvian adult population [26].

Gender	Average Intake by Different Age Groups (g day^−1^)			
	19–34	35–49	60–64	All
Male	105.0	88.0	66.6	85.2
Female	59.1	58.0	57.6	58.1

**Table 4 toxins-16-00053-t004:** PDI values for males and females based on the average UB mycotoxin contamination levels in the studies of plant-based beverages.

Mycotoxin	PDI (Males, UB) (ng kg^−1^ Body Weight per Day)				PDI (Females, UB) (ng kg^−1^ Body Weight per Day)			
	Age (Years)							
	19–34	35–49	50–64	All	19–34	35–49	50–64	All
DON	1.87	1.57	1.19	1.52	1.05	1.03	1.03	1.04
ZEN	0.34	0.28	0.22	0.28	0.19	0.19	0.19	0.19
OTA	0.19	0.16	0.12	0.15	0.10	0.10	0.10	0.10
NIV	7.62	6.39	4.84	6.19	4.29	4.21	4.18	4.22
Sum T-2 + HT-2	1.69	1.41	1.07	1.37	0.95	0.93	0.92	0.93
AFB1	0.08	0.07	0.05	0.07	0.05	0.05	0.05	0.05
AFB1 + AFB2	0.10	0.09	0.07	0.08	0.06	0.06	0.06	0.06
AOH	1.34	1.12	0.85	1.09	0.75	0.74	0.74	0.74
AME	1.08	0.91	0.69	0.88	0.61	0.60	0.59	0.60
TEN	0.62	0.52	0.39	0.50	0.35	0.34	0.34	0.34
BEA	0.24	0.20	0.15	0.19	0.13	0.13	0.13	0.13
ENNs	1.57	1.32	1.00	1.28	0.89	0.87	0.86	0.87

**Table 5 toxins-16-00053-t005:** Exposure assessment for males and females based on the determined average UB mycotoxin contamination levels in plant-based milk substitute beverages.

Mycotoxin	%TDI (Male, UB) %				%TDI (Female, UB) %			
	Age (Years)							
	19–34	35–49	50–64	All	19–34	35–49	50–64	All
DON	0.19	0.16	0.12	0.15	0.11	0.10	0.10	0.10
ZEN	0.14	0.11	0.09	0.11	0.08	0.08	0.07	0.08
OTA	6.21	5.20	3.94	5.04	3.50	3.43	3.41	3.44
NIV	1.91	1.60	1.21	1.55	1.07	1.05	1.05	1.05
Sum T-2/HT-2	8.43	7.06	5.35	6.84	4.74	4.65	4.62	4.66
AFB1	8.25	6.91	5.23	6.69	4.64	4.56	4.53	4.57
AFB2	2.19	1.84	1.39	1.78	1.23	1.21	1.20	1.21
AFB1 + AFB2	10.44	8.75	6.62	8.47	5.88	5.77	5.73	5.78
AOH	53.62	44.94	34.01	43.51	30.18	29.62	29.41	29.67
AME	43.20	36.21	27.40	35.05	24.32	23.86	23.70	23.90
TEN	0.04	0.03	0.03	0.03	0.02	0.02	0.02	0.02
BEA	0.95	0.80	0.60	0.77	0.53	0.52	0.52	0.53
ENNs	0.16	0.13	0.10	0.13	0.09	0.09	0.09	0.09

**Table 6 toxins-16-00053-t006:** PDI values for males and females based on the maximum determined mycotoxin contamination levels in the studies of plant-based beverages.

Mycotoxin	PDI (Males, UB) (ng kg^−1^ Body Weight per Day)				PDI (Females, UB) (ng kg^−1^ Body Weight per Day)			
	Age (Years)							
	19–34	35–49	50–64	All	19–34	35–49	50–64	All
DON	12.9	10.8	8.2	10.5	7.3	7.1	7.1	7.1
ZEN	0.44	0.36	0.28	0.35	0.24	0.24	0.24	0.24
OTA	0.90	0.75	0.57	0.73	0.51	0.50	0.49	0.50
NIV	8.10	6.79	5.14	6.57	4.56	4.47	4.44	4.48
Sum T-2 + HT-2	4.80	4.02	3.04	3.89	2.70	2.65	2.63	2.66
AFB1	0.11	0.09	0.07	0.09	0.06	0.06	0.06	0.06
AFB1 + AFB2	0.03	0.03	0.02	0.02	0.02	0.02	0.02	0.02
AOH	0.14	0.11	0.09	0.11	0.08	0.07	0.07	0.07
AME	3.00	2.51	1.90	2.43	1.69	1.66	1.65	1.66
TEN	4.50	3.77	2.85	3.65	2.53	2.49	2.47	2.49
BEA	4.05	3.39	2.57	3.29	2.28	2.24	2.22	2.24
ENNs	0.99	0.83	0.63	0.80	0.56	0.55	0.54	0.55

**Table 7 toxins-16-00053-t007:** Exposure assessment for males and females based on the determined maximum determined mycotoxin contamination levels in plant-based milk substitute beverages.

Mycotoxin	%TDI (Male, Max) %				%TDI (Female, Max) %			
	Age (Years)							
	19–34	35–49	50–64	All	19–34	35–49	50–64	All
DON	1.29	1.08	0.82	1.05	0.73	0.71	0.71	0.71
ZEN	0.17	0.15	0.11	0.14	0.10	0.10	0.10	0.10
OTA	30.00	25.14	19.03	24.34	16.89	16.57	16.46	16.60
NIV	2.03	1.70	1.28	1.64	1.14	1.12	1.11	1.12
Sum T-2/HT-2	24.00	20.11	15.22	19.47	13.51	13.26	13.17	13.28
AFB1	10.50	8.80	6.66	8.52	5.91	5.80	5.76	5.81
AFB2	3.00	2.51	1.90	2.43	1.69	1.66	1.65	1.66
AFB1 + AFB2	13.50	11.31	8.56	10.95	7.60	7.46	7.41	7.47
AOH	120.00	100.57	76.11	97.37	67.54	66.29	65.83	66.40
AME	180.00	150.86	114.17	146.06	101.31	99.43	98.74	99.60
TEN	0.27	0.23	0.17	0.22	0.15	0.15	0.15	0.15
BEA	3.96	3.32	2.51	3.21	2.23	2.19	2.17	2.19
ENNs	0.80	0.67	0.51	0.65	0.45	0.44	0.44	0.44

**Table 8 toxins-16-00053-t008:** List of analytes and precursor and fragment ions (quantifier ions are in bold).

Compound	RT, min	Acquisition Window, min	Polarity	Precursor *m*/*z*	Product *m*/*z*	CE, V	Min Dwell Time (ms)
DON	2.1	4	+	297	**231**	12	9.278
					249	11	9.278
3-ADON and 15-ADON	5.4	6	+	339	**137**	15	3.676
					321	5	3.676
					231	13	3.676
ZEA	7.0	5	-	317	131	28	3.676
					**175**	24	3.676
					273	18	3.676
FB1	6.5	5	+	723	**334**	40	3.676
					352	35	3.676
FB2	6.8	5	+	707	**318**	38	3.676
					336	36	3.676
					354	33	3.676
OTA	7.0	5	+	404	102	50	3.676
					**239**	25	3.676
					358	13	3.676
NIV	1.3	2	+	313	**125**	16	48.28
					175	13	48.28
T-2	6.8	5	+	489	**327**	23	3.676
					387	21	3.676
HT-2	6.5	5	+	442	**197**	16	3.676
					263	10	3.676
AFB1	6.1	5	+	313	**213**	43	3.676
					241	37	3.676
					285	23	3.676
AFB2	6.0	5	+	315	**259**	33	3.676
					287	24	3.676
AFG1	5.9	5	+	329	**200**	44	3.676
					243	28	3.676
					283	26	3.676
AFG2	5.7	5	+	331	**189**	51	3.676
					245	32	3.676
					275	29	3.676
AOH	6.5	5	-	257	**147**	25	3.676
					213	16	3.676
AME	7.2	5	-	271	**227**	39	3.676
					256	16	3.676
TEN	6.6	5	+	416	**199**	10	3.676
					256	30	3.676
					312	20	3.676
BEA	8.0	5	+	784	134	53	3.676
					**244**	26	3.676
ENNA	8.1	5	+	685	**210**	31	3.676
					228	33	3.676
ENNA1	8.3	5	+	699	**210**	34	3.676
					228	34	3.676
ENNB	7.9	5	+	657	**196**	32	3.676
					214	34	3.676
ENNB1	8.0	5	+	671	**196**	34	3.676
					214	35	3.676

**Table 9 toxins-16-00053-t009:** Parameters used for the calculation of PDI.

Mycotoxin	TDI (ng kg^−1^ Body Weight per Day)	TTC (ng kg^−1^ Body Weight per Day)	Reference
DON	1000 *	-	[32]
ZEA	250 *	-	[35]
NIV	400 *	-	[33]
SUM T-2/HT-2	20 *	-	[34]
SUM ENNs	-	1500	[36]
BEA	-	25	[36]
OTA	3 *	-	[39]
AFB1	-	1 **	[38]
SUM of AFs	-	1 **	[38]
AOH	-	2.5	[37]
AME	-	2.5	[37]
TEN	-	1500	[37]

* European Food Safety Authority (EFSA), ** CPF—cancer potency factor.

## Data Availability

All data are present in the article.

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
