# Peer review of "An Occurrence Study of Mycotoxins in Plant-Based Beverages Using Liquid Chromatography–Mass Spectrometry"

_toxins, 2024, doi:10.3390/toxins16010053_

Round 1

Reviewer 1 Report

Comments and Suggestions for Authors

Review report “toxins-2786323”

In the manuscript entitled “Occurrence study of mycotoxins in plant-based beverages using liquid chromatography-mass spectrometry” the authors report on the monitoring of 16 mycotoxins in 72 samples of plant-based beverages, then they tested their thermal stability in the matrices and performed a risk assessment study.

The manuscript is quite well written and the length is appropriate. The abstract is quite clear and sufficiently reflects the manuscript content. There are some grammar errors, so the authors should implement this part (e.g. lines 35, 101, 102…).

I have some suggestion and comments for the authors:

Results and discussion

Risk assessment is a very structured process and exposure assessment is a part of it, so the term “exposure risk assessment”, used in discussion and tables, is wrong. I suggest the authors to re-check the terminology (use the EFSA glossary!).

About method optimization, the authors described extraction optimization, but the number of replicates and the statistical analysis were not reported. Similarly for the validation study part 2.2 the data reported are summary and often mis-correct. The validation study reported by the authors is almost an act of faith.

This part is not organized in a logical way. Moreover, several sections are not necessary (for example all the equations in the paragraph 2.2.2 and 2.2.3) and some others should be moved in M&M section. In the “Results and Discussion” you should report exactly “results and discussion” not how you calculated the validation parameters.

Matrix effect calculation how many replicates of calibration curve??

M&M

- Add (Company, City, Country) for all instrumentations reagents standards, software. Standardize it

- line 552 add reference

- line 560 re-write the sentence

- line 496 How many replicates for each analysis?

Based on these comments I strongly encourage the authors to improve the manuscript, since, in its current form, it is not eligible for publication in Toxins.

Comments on the Quality of English Language

Moderate editing of English language is required

Author Response

Response to Reviewer 1 Comments

1. Summary

Thank you very much for taking the time to review this manuscript. Please find the detailed responses below and the corresponding revisions and in track changes in the re-submitted files.

2. Questions for General Evaluation

Reviewer’s Evaluation

Response and Revisions

Does the introduction provide sufficient background and include all relevant references?

Yes

Are all the cited references relevant to the research?

Can be improved

We have changed the references.

Is the research design appropriate?

Must be improved

Improvements have been made. The explanation is described in the 3rd Point-by-point response

Are the methods adequately described?

Must be improved

Improvements have been made. The explanation is described in the 3rd Point-by-point response

Are the results clearly presented?

Can be improved

Improvements have been made. The explanation is described in the 3rd Point-by-point response

Are the conclusions supported by the results?

Can be improved

Improvements have been made. The explanation is described in the 3rd Point-by-point response

3. Point-by-point response to Comments and Suggestions for Authors

Comments 1: The manuscript is quite well written and the length is appropriate. The abstract is quite clear and sufficiently reflects the manuscript content. There are some grammar errors, so the authors should implement this part (e.g. lines 35, 101, 102…).

Response 1: Thank you for pointing this out. The errors you highlighted have been corrected (Page 3, 2.1.1. Comparison of the SPE and QuEChERS methodologies, Line 149 – 150).

Line 28: Consumers choose these beverages as alternatives to dairy milk due to health, environmental, and lifestyle preference reasons.

Line 137-138 (previous lines 101-102): “For most of the mycotoxin, the EE was higher using the QuEChERS methodology, with the exception of HT-2 toxin and fumonisin B1, for which efficiency was higher in the case of SPE procedure.”

Comments 2: Risk assessment is a very structured process and exposure assessment is a part of it, so the term “exposure risk assessment”, used in discussion and tables, is wrong. I suggest the authors to re-check the terminology (use the EFSA glossary!).

Response 2: Thank you for your comment, we have changed the sentences in the manuscript to “Exposure assessment”.

Comments 3: About method optimization, the authors described extraction optimization, but the number of replicates and the statistical analysis were not reported. Similarly for the validation study part 2.2 the data reported are summary and often mis-correct. The validation study reported by the authors is almost an act of faith.

Response 3: Thank you for your comment. Regarding the method validation part, we should note that all parameters were validated in five replicates (Aflatoxins in 3 replicates) (Page 4, 2.2. Validation of the Method, Line 228 – 229).

“The method characteristics including recovery, repeatability, selectivity, linearity, method sensitivity, and matrix effects were evaluated (see Table S1).”

In relation to the method optimization part, we have corrected this mistake and indicated that all experiments were conducted in three replicates.

Thank you for your comment about the insufficient level of validation, we still believe that validation is done to the sufficient extent, and all validation data is summarized in Table 2, and Table S1. All parameters were tested in replicates, for each matrix.

Comments 4: This part is not organized in a logical way. Moreover, several sections are not necessary (for example all the equations in the paragraph 2.2.2 and 2.2.3) and some others should be moved in M&M section. In the “Results and Discussion” you should report exactly “results and discussion” not how you calculated the validation parameters.

Response 4: We agree with your comment, the equations have been transferred to the M&M section (Page 15 – 16, 4.4. Extraction Efficiency, Repeatability, Recovery and Matrix Effect, Line 784 – 807).

Comments 5: Matrix effect calculation how many replicates of calibration curve??

Response 5: During testing the matrix effect, we did not have replicates because we used the calibration curve for each of the four matrices. Each calibration curve had 3-5 calibration points for each analyte and for each matrix. To quantify mycotoxin in each sequence, we used a calibration curve.

Comments 6: M&MAdd (Company, City, Country) for all instrumentations reagents standards, software. Standardize it.

Response 6: Thank you for pointing this out, we have rewritten the Materials and Methods part.

Comments 7: M&M – line 552 add reference.

Response 7: The reference European Food Safety Authority. Management of left‐censored data in dietary exposure assessment of chemical substances. EFSA J. 2010, 8(3), 1557. [29] was already added to the sentence.

Comments 8: M&M – line 560 re-write the sentence.

Response 8: The sentence was rewritten: A recent study reported the cancer potency factor (CPF) proposed by JECFA, which was set in the case of the dietary exposure level of AFB1 at 1 ng μg kg−1 bw per day.

Comments 9: M&M – line 496 How many replicates for each analysis?

Response 9: Thank you for the question, the single measurement was done for each sample if they did not exceed the last calibration point. According to EU Commission Regulation No 401/2006, repeating the sample is not necessary if the mycotoxin concentration does not exceed 50% of the maximum level.

4. Response to Comments on the Quality of English Language

Point 1: Moderate editing of English language is required

Response 1: Thank you for your comment about English language. We corrected the English in the manuscript and answered the reviewer's questions.

5. Additional clarifications

We do not have any additional clarifications.

Reviewer 2 Report

Comments and Suggestions for Authors

Comments on the Quality of English Language

Some minor comment has been included in the word document

Author Response

Response to Reviewer 2 Comments

1. Summary

Thank you very much for taking the time to review this manuscript. Please find the detailed responses below and the corresponding revisions and in track changes in the re-submitted files.

2. Questions for General Evaluation

Reviewer’s Evaluation

Response and Revisions

Does the introduction provide sufficient background and include all relevant references?

Can be improved

Authors have provided corrections. Response in the 3. point-by-point response.

Are all the cited references relevant to the research?

Yes

Is the research design appropriate?

Yes

Are the methods adequately described?

Can be improved

Authors have provided corrections. Response in the 3. point-by-point response.

Are the results clearly presented?

Yes

Are the conclusions supported by the results?

Yes

3. Point-by-point response to Comments and Suggestions for Authors

Comments 1: Although authors evaluate all the points expected to be covered in such a study, in my opinion the work does not show its novelty. Indeed, authors mentioned some articles already developed to analyze mycotoxins in similar matrices, without highlighting the novelty of its own work.

Response 1: Thank you for pointing this out. To show our research’s novelty, we added some text to the introduction (Page 2, 1. Introduction, Line 116 – 119).

In this study, low quantification levels were achieved for this type of matrix, new compounds (NIV, TEN, 3-ADON and 15-ADON) not previously studied in plant-based beverages were investigated as well as new matrices such as hemp, millet, buckwheat and pea beverages have been included in the study.

Comments 2: Abstract: You may include some information about the exposure risk assessment study.

Response 1: The manuscript was corrected. The sentence “European Commission has not yet set guidelines for the maximum mycotoxin concentrations in plant-based beverages, nor has the European Food Safety Authority conducted a risk assessment. Therefore, acute exposure studies were provided for the Latvian population based on the assumed replacement of dairy milk with plant-based beverages to ascertain the safety of plant-based milk substitutes” provides the information about the conducted exposure assessment.

Comments 3: Introduction: It is true there is no actual regulation for mycotoxins in plant-based drinks. However, you may cite maximum residue limits (MRLs) for similar food commodities. For example, the legal limits for aflatoxin M1 in milk, or the MRLs for several mycotoxins in cereals and nuts proposed by the European Union regulation.

Response 3: Thank you for pointing this out. We have, accordingly, modified the Introduction part to emphasize this point (Page 2, 1. Introduction, Line 96 – 107).

The European Commission Regulation No 2023/915 sets maximum levels for the main mycotoxins for similar food commodities, and the Commission Recommendation No 2013/165 sets the indicative levels for T-2 and HT-2 toxin in cereals and cereal products [18,19]. Indicative levels of 200 and 50 μg kg-1 are recommended for oat bran and flakes, and other cereal milling products for the sum of T-2 and HT-2 toxins. For OTA, DON, ZEA, and AFB1 maximum levels of 3.0 μg kg-1, 200 μg kg-1, 20 μg kg-1, and 2.0 μg kg-1, respectively, are indicated for products derived from unprocessed cereal grains and processed cereal foods. The maximum levels for AFB1 and the sum of aflatoxins in oil seeds (such as soybeans, palm nuts, and hemp seeds) and tree nuts (including cashew nuts and cocoa beans) are 2.0 μg kg-1 and 4.0 μg kg-1, respectively. For almonds, hazelnuts, and Brazil nuts intended for use as ingredients in food, the maximum levels are between μg kg-1 and 8.0 μg kg-1 for AFB1 [18,19].

Comments 4: Line 41-44: Although it is true mycotoxins could be found in raw ingredients as you commented, they may not necessarily be transferred to plant-based drinks. You could include reference 14-16 here to assess it.

Response 4: We agree with this comment. Therefore, we have added references (Page 2, 1. Introduction, Line 72).

“Raw agricultural products, including cereals, legumes, nuts, fruits, herbs, and other crops used in herbal beverages, can be contaminated with fungi leading to the occurrence of mycotoxins in processed foods such as plant-based beverages [9–11].”

Comments 5: Line 80: A space was missing in “(QuEChERS)method”.

Response 5: Thank you, we agree with this comment. We have modified the sentence to correct this point (Page 3, 2.1.1. Comparison of the SPE and QuEChERS methodologies, Line 135).

”Two sample preparation methods were compared, namely, “quick, easy, cheap, effective, rugged, and safe” (QuEChERS) method and C18 column based solid phase extraction (SPE) based on the evaluation of the extraction efficiency (EE).”

Comments 6: Line 86: Do you mean the area of the peak for the extracted ion chromatogram of each compound?

Response 6: Thank you for your question. Yes, we mean the area of the peak for the extracted ion chromatogram of each compound. We have added some text to emphasize this point (Page 15, 4.4. Extraction Efficiency, Repeatability, Recovery and Matrix Effect, Line 784 – 786).

“The EE was tested as the ratio between the blank sample spiked with standard after the procedure and the sample spiked with standard at the beginning of the procedure for each compound (equation 1)…”

Comments 7: Formula (1): To indicate % you need to multiply the ratio by 100.

Response 7: I agree with this comment. We have changed the formula (Page 15, 4.4. Extraction Efficiency, Repeatability, Recovery and Matrix Effect, Equation 1).

Comments 8: The meaning of Aafter is missing.

Response 8: Thank you for this comment, we have added this part to the text (Page 15, 4.4. Extraction Efficiency, Repeatability, Recovery and Matrix Effect, Line 790 – 791).

“Where Abefore is the area of the chromatographic peak for a spiked sample before the extraction procedure, Anon spiked is the area of the chromatographic peak for a blank sample at the end of sample preparation procedure, Aafter is the area of the chromatographic peak for blank sample spiked with the standard after sample preparation.”

Comments 9: Line 89: If the method is performed in liquid samples, validation concentrations are in μg L-1 (e.g. 0.031 μg L-1).

Response 9: We did not use ”μg L-1” because all the samples were weighed on the scales, and all standard additions were added as “μg kg-1”. Additionally, each plant-based beverage has a different density, which means that “μg L-1” is not equivalent to “μg kg-1”.

Comments 10: Line 102: Change “fumonisin B1 were higher efficiency was determined in the case of SPE procedure” by “fumonisin B1, for which efficiency were higher in the case of SPE procedure”.

Response 10: Authors agree. We have, accordingly, changed the sentence to correct this point (Page 3, 2.1.1. Comparison of the SPE and QuEChERS methodologies, Line 150).

“…with the exception of HT-2 toxin and fumonisin B1, for which efficiency was higher in the case of SPE procedure.”

Comments 11: Figure 1: You should provide an explanation for the low recoveries obtained for BEA and enniatins when using QuEChERS extraction.

Response 11: Thank you for pointing this out, we have explained the low recoveries for BEA and enniatins (Page 3, 2.1.1. Comparison of the SPE and QuEChERS methodologies, Line 157 – 158).

The QuEChERS method resulted in low analyte signals for BEA and ENN group, because of inefficient extraction to the organic layer [21].

Comments 12: Line 110: For QuEChERS extraction? Clarify.

Response 12: Thank you for this question; yes, it means optimizing the sample preparation for QuEChERS extraction (Page 3, 2.1.1. Comparison of the SPE and QuEChERS methodologies, Line 159).

“During the sample preparation optimization for QuEChERS extraction, extraction efficiencies were…”

Comments 13: Line 137: What is the purpose of using ammonium acetate and formic acid together? If it is to obtain a buffer effect, ammonium acetate and acetic acid or ammonium formate and formic acid should be used. Please, explain it. The same comment for mobile phases in lines 503-504.

Response 13: Thank you for pointing this out. We used a combination of ammonium acetate and formic acid as mobile phases and for reconstruction to improve the ionization efficiency. We chose this combination based on its common usage in similar studies (see references below). All of the publications also used ammonium acetate and formic acid:

1. Zadeike, D.; Vaitkeviciene, R.; Bartkevics, V.; Bogdanova, E.; Bartkiene, E.; Lele, V.; Juodeikiene, G.; Cernauskas, D.; Valatkeviciene, Z. The Expedient Application of Microbial Fermentation after Whole-Wheat Milling and Fractionation to Mitigate Mycotoxins in Wheat-Based Products. LWT, 2021, 137, 110440. https://doi.org/10.1016/j.lwt.2020.110440.

2. Janaviciene, S.; Suproniene, S.; Kadziene, G.; Pavlenko, R.; Berzina, Z.; Bartkevics, V. Toxigenicity of F. Graminearum Residing on Host Plants Alternative to Wheat as Influenced by Environmental Conditions. Toxins, 2022, 14, 541. https://doi.org/10.3390/toxins14080541.

3. Janaviciene, S.; Venslovas, E.; Kadziene, G.; Matelioniene, N.; Berzina, Z.; Bartkevics, V.; Suproniene, S. Diversity of Mycotoxins Produced by Fusarium Strains Infecting Weeds. Toxins, 2023, 15, 420. https://doi.org/10.3390/toxins15070420.

Comments 14: Line 150: Tell the injection volumes tested.

Response 14: The sentence was amended (Page 4, 2.1.2. Selection of a Reconstitution Solution and Sample Filtration, Line 217).

“The selected reconstitution solution did not match the initial mobile phase composition, therefore the injection volumes were tested, and 5 µL was chosen to achieve the optimal signal shape for DON and NIV.”

Comments 15: Line 164: Eliminate a “,”.

Response 15: Thank you for pointing this out, we have corrected this error (Page 4, 2.2.1. Levels of Quantification, Method Selectivity, and Linearity, Line 230).

“2.2.1. Levels of Quantification, Method Selectivity, and Linearity”

Comments 16: Line 166: Did you test it in different blanks samples? Specify in what samples you have tested.

Response 16: Thank you for your question. Yes, in the publication, we tested four different blank samples: soy drink, oat drink, rice drink, and nut drink (Page 4, 2.2. Validation of the Method, Line 224).

“…the method was validated using four different types of plant-based (e.g., oat, rice, soya, and almond) beverages…”

Comments 17: Line 171: Indicate the concentrations of the procedural calibration points.

Response 17: Thank you for pointing this out. We have made some changes in this part of the text. The concentrations of the procedural calibration points are from the LOQ level to 40xLOQ level (Page 4, 2.2.1. Levels of Quantification, Method Selectivity, and Linearity, Line 237).

“The linearity and correlation coefficients (R2 > 0.98) of the method were tested by a 5-point procedural calibration ranging from the LOQ level to 40×LOQ level.”

Comments 18: Line 174: Although normally S/N>10 is accepted to establish the LOQs, it is recommended that accuracy and precision values are acceptable (recoveries 70-120% and RSD<20%), as indicated in the latest version of SANTE guideline.

Response 18: Thank you for your comment, for mycotoxin validation we did not use pesticide SANTE guideline. In our study, we applied EU Commission Regulation No 401/2006 and EURL Guidance Document on the Estimation of LOD and LOQ for Measurements in the Field of Contaminants in Food and Feed. Our LOQ levels are determined by testing recovery and RSD in the validation section.

Comments 19: Table 2: I think it could be more adequate to indicate that n=3 as table foot or in the table title.

Response 19: We agree with you; so, we have changed the table title (Page 5, 2.2.2. Repeatability and Recovery, Line 272).

Table 2. The average recovery values for mycotoxins (n = 3, test repetitions)”

Comments 20: Line 213: this is not exactly like this. If ME is between -20 and 20%, it is considered that there is no matrix effect. On the other hand, ME<-20% indicate ion suppression, and ME>20% ion enhancement.

Response 20: Thank you so much for this comment, so, we agree with you. We have changed the description about the matrix effect (Page 16, 4.4. Extraction Efficiency, Repeatability, Recovery and Matrix Effect, Line 804 – 806).

“If the matrix effect is between -20% and 20%, it is considered that this effect is negligible. The matrix effect that is lower than -20% corresponds with an ion suppression, while higher than 20% corresponds with an ion enhancement.”

Comments 21: Line 226: For those compounds in which recoveries were lower than 70%, you could mention if RSD were lower than 20%. In these cases, a correction factor could be applied for quantification purposes.

Response 21: Thank you for pointing this out, but in line 226, we discuss extraction efficiency, which is not the same as recovery. Therefore, we did not mention RSD. The EU Commission Regulation No 401/2006 does not mention the use of the correction factor if the extraction efficiency is lower than 70%.

Comments 22: Figure 3: How do you explain the increase of ZEA after boiling?

Response 22: Thank you for the question, the only explanation currently we have is related to measurement uncertainty and inhomogeneity of samples.

Comments 23: Section 2.4. Results of Thermal Stability Testing: You must present the results evaluating the statistically changes during heating to assure if mycotoxins significantly decreased after these treatments.

Response 23: Thank you for pointing this out, but as you can see in Figure 3 and 4, we displayed the standard deviation of each result.

Comments 24: Line 306: Why is reference 25 in red?

Response 24: Thank you for pointing this, the unintentional error occurred during the publication's modification (Page 8, 2.5.1. Consumption of Milk in Latvia, Line 540).

“…for different milk products including pasteurised dairy milk among adults in Latvia [26].”

Comments 25: Table 3: One column is not centered.

Response 25: We agree with your comment, thank you, we have changed the design of table (Page 8 – 9, 2.5.1. Consumption of Milk in Latvia, Table 3).

Comments 26: Line 426: What requirements? Do you mean the maximum residue limits?

Response 26: Thank you for pointing this, we have edited this part of the text (Page 12, 3. Conclusions, Line 675 – 676). It means that the method parameters complied with EU Commission Regulation.

“…complied with the European Union Commission Regulation No 401/2006 [24].

Comments 27: Line 498: Did you optimize different chromatographic conditions (gradient, mobile phases, flows, etc)? If yes, please specify it.

Response 27: Thank you for your question. Yes, we tried different chromatographic columns (Kinetex Biphenyl 50 x 2.1 mm, 1.7 μm; Luna Omega Polar 50 x 2.1 mm, 1.6 μm; Kinetex C18 50 x 3 mm, 1.7 μm), but this information is not shown in the publication.

4. Response to Comments on the Quality of English Language

Point 1: Minor editing of English language required.

Response 1: Thank you for your comment about the English language, in the manuscript was rechecked

5. Additional clarifications

We do not have any additional clarifications.

Round 2

Reviewer 1 Report

Comments and Suggestions for Authors

Thank you for the work of revision done in the manuscript. Here some comments:

-        Again, I feel forced to specify to the authors that a good validation study does not consist in report the formulas/equations of some parameters in the M&M and absolutely not in results…. You should simplify these parts.

-        Moreover, in table S1, a value of uncertainty for each analyte is reported. How do you estimate this fundamental parameter?????

-        I do not understand the rationale for the choice of measurement number for estimation of recovery and reproducibility; from the EuraChem guide The Fitness for Purpose of Analytical Methods “The minimum number of replicates specified varies with different protocols but is typically between 6 and 15 for each material used in the study”. Could you explain me your choice of very different number of measurements among analytes/concentrations and matrices?

-        The authors stated “the method repeatability and recovery were tested for each matrix sample spiked with a mycotoxin standard at the LOQ level and at 2-5 times the LOQ level.”

Maybe, looking at table S1 is better (and even more logical) to say 2-10 times.

Please clarify these points.

Author Response

For research article

Response to Reviewer 1 Comments

1. Summary

Thank you very much for taking the time to review this manuscript. Please find the detailed responses below and the corresponding revisions and in track changes in the re-submitted files.

2. Questions for General Evaluation

Reviewer’s Evaluation

Response and Revisions

Does the introduction provide sufficient background and include all relevant references?

Yes

Are all the cited references relevant to the research?

Can be improved

We have changed the references.

Is the research design appropriate?

Can be improved

Improvements have been made. The explanation is described in the 3rd Point-by-point response

Are the methods adequately described?

Can be improved

Improvements have been made. The explanation is described in the 3rd Point-by-point response

Are the results clearly presented?

Can be improved

Improvements have been made.

Are the conclusions supported by the results?

Can be improved

Improvements have been made.

3. Point-by-point response to Comments and Suggestions for Authors

Comments 1: Again, I feel forced to specify to the authors that a good validation study does not consist in report the formulas/equations of some parameters in the M&M and absolutely not in results…. You should simplify these parts.

Response 1: Thank you for pointing this out. We have deleted all equations from the Result part, and some equations from the M&M part. Still we consider that equations explaining calculations of the matrix effect and extraction efficiency should be shown in the M&M part since scientific publications have different ways of expressing these parameters.

Comments 2: Moreover, in table S1, a value of uncertainty for each analyte is reported. How do you estimate this fundamental parameter?????

Response 2: Thank you for your comment, we have calculated the uncertainty using the equation shown below [1, 2]:

, where R is recovery (%), and RSD is relative standard deviation (%).

[1] Reinholds, I., Rusko, J., Pugajeva, I., Berzina, Z., Jansons, M., Kirilina-Gutmane, O., Tihomirova, K., & Bartkevics, V. (2020). The Occurrence and Dietary Exposure Assessment of Mycotoxins, Biogenic Amines, and Heavy Metals in Mould-Ripened Blue Cheeses. In Foods (Vol. 9, Issue 1, p. 93). MDPI AG. https://doi.org/10.3390/foods9010093

[2] Karapinar, H. S., Eczacioglu, N., & Dogan, F. (2024). Comprehensive and sensitive validation of the method and determination of measurement uncertainty for simultaneous specification of aflatoxin B1, B2, G1 and G2 in nuts. In Measurement: Food (Vol. 13, p. 100124). Elsevier BV. https://doi.org/10.1016/j.meafoo.2023.100124

Comments 3: I do not understand the rationale for the choice of measurement number for estimation of recovery and reproducibility; from the EuraChem guide The Fitness for Purpose of Analytical Methods “The minimum number of replicates specified varies with different protocols but is typically between 6 and 15 for each material used in the study”. Could you explain me your choice of very different number of measurements among analytes/concentrations and matrices?

Response 3: Thank you for your comment.

Thank you for your suggestions, we agree that the initial version of the manuscript ambiguously presented the validation design.

In the validation protocol for all beverages, two concentration levels were tested (LOQ and about ) in five replicates for each level (only for aflatoxins the number of replicates is different, because four aflatoxins were in the mixture). In addition to that we used data from quality control experiments (spiked samples) for calculation of method performance characteristics (for example, uncertainty)

Now we have made the description of validation experiments more comprehensive.

Comments 4: The authors stated “the method repeatability and recovery were tested for each matrix sample spiked with a mycotoxin standard at the LOQ level and at 2-5 times the LOQ level.” Maybe, looking at table S1 is better (and even more logical) to say 2-10 times.

Response 4: We agree with your comment, we have rewritten the sentence (Page 5, 2.2.2. Repeatability and Recovery, Line 210).

The method repeatability and recovery were tested for each matrix sample spiked with a mycotoxin standard at the LOQ level and at 2-10 times the LOQ level.

4. Response to Comments on the Quality of English Language

Point 1: English language fine. No issues detected

Response 1: Thank you for your comment about English language.

5. Additional clarifications

We do not have any additional clarifications.

Round 3

Reviewer 1 Report

Comments and Suggestions for Authors

The authors addressed reviewer's comments.